# A Fast Transient Adaptive On-Time Controlled BUCK Converter with Dual Modulation

**DOI:** 10.3390/mi14101868

**Published:** 2023-09-29

**Authors:** Mengyuan Sun, Chufan Chen, Leiyi Wang, Xinling Xie, Yuhang Wang, Min Xu

**Affiliations:** 1State Key Laboratory of ASIC and System, School of Microelectronics, Fudan University, Shanghai 200433, China; 20212020003@fudan.edu.cn (M.S.); 21212020002@m.fudan.edu.cn (C.C.); lywang2018@stu.suda.edu.cn (L.W.); 21212020013@m.fudan.edu.cn (X.X.); yuhangwang21@m.fudan.edu.cn (Y.W.); 2Shanghai Integrated Circuit Manufacturing Innovation Center Co., Ltd., Shanghai 200433, China

**Keywords:** DC-DC, buck converter, adaptive on-time, fast response, constant frequency, pulse skip modulation

## Abstract

This paper proposed a fully integrated adaptive on-time (AOT) controlled buck converter with fast load transient. An adaptive on-time generator is presented to stabilize the output frequency. To enhance the light load efficiency, the converter could transfer from the pulse width modulation (PWM) to pulse skip modulation (PSM) as the load current decreases. The buck converter can switch between these two modulation modes adaptively with the assistance of a zero current detection circuit. Implemented in the TSMC 0.18 µm BCD (BiCMOS/DMOS) process, the proposed buck converter works with an input voltage ranging from 5.5 to 15 V, an output voltage ranging from 0.5 to 5 V, and an output load ranging up to 5 A. The experimental results show that based on the dual modulation adaptive on-time controlled mode, the transient recovery time from light to heavy load and from heavy load to light load is 13 µs and 15 µs, respectively. An overshot voltage of 57 mV and an undershot voltage of 53 mV are also achieved.

## 1. Introduction

As communication and semiconductor technologies continue to advance, electronic equipment is also evolving rapidly and becoming more intelligent. As electronic systems grow more complex, the power supply modules have played an increasingly critical role in these systems [1,2]. The DC-DC power management chip is indispensable for electronic equipment to ensure the stability of the power supply and reduce system power consumption [3]. The DC-DC converter chip adjusts itself by monitoring the power output energy and response for the transformation, distribution, detection, and management of electrical energy in the system so that the battery power can be reasonably distributed to different modules in the system to save energy and ensure stable power supply [4,5].

The power management chips for microprocessors are fast-changing loads, so the power management chip needs to respond to load changes quickly [6,7]. With the increasing demand for intelligence and modularity, the load will change to be faster and more complex, which puts higher demands on the dynamic characteristics of the power management chip [8,9,10]. Feedback control is the main factor to ensure the converter’s output is stable under different load conditions. The approaches for feedback control are generally voltage mode and current mode [11,12]. The buck converter with voltage mode control has a simpler structure but has more complex compensation circuits and slower transient response, and it is unable to monitor current [13]. The constant on-time (COT) controlled buck converters have recently been proposed due to their simple compensation structure and fast load transient response [14]. However, subharmonic oscillations may occur in some cases [15,16]. In this paper, a dual-loop feedback control is proposed to enhance the response speed while ensuring loop stability [17].

Since the on-time of power transistors per cycle is kept constant in COT control, the duty cycle is fixed for a defined input and output voltage. Thus, the operating frequency of the converter will vary with different input and output voltages [18]. However, once the switching frequency of the converter changes with the working conditions, it will bring serious electromagnetic interference (EMI) problems, and the converter will be restricted to applications that require fixed frequency. The buck converter proposed in this paper used an adaptive on-time generator to produce the on time that varies with the working conditions, which can keep the system operating at a constant frequency. In order to improve the system efficiency under a wide load range, pulse width modulation (PWM) combined with pulse skip modulation (PSM) has been used [19]. Under heavy load conditions, the main loss of the system is the conduction loss due to the load current going through the power transistors, and the PWM control can obtain high efficiency. The switching loss under light load is mainly caused by the charging and discharging of the gate capacitor of power transistors. The light load power consumption could be reduced by reducing the operating frequency to adopt PSM control [20]. The proposed zero-current detection (ZCD) circuit enables the converter to switch smoothly between two modulation modes.

This paper presents a fast adaptive on-time controlled buck converter with a dual feedback control loop compared with voltage-controlled and current-controlled modes. It can achieve a higher response speed, and compared with the traditional constant-on-time controlled mode, it can stabilize the output frequency. This converter could work under PWM control, which keeps the operating frequency stable and reduces the adverse effects of frequency changes. Furthermore, the PSM can be used under light loads to improve efficiency. The rest of this paper is organized as follows. The system of the proposed buck converter is described in Section 2. Section 3 illustrates the circuit-level implementations of the adaptive on-time generator and the zero-current detection circuit. The experimental results are presented in Section 4. Finally, Section 5 concludes this paper.

## 2. Proposed Fast Transient AOT-Controlled Buck Converter

Figure 1 illustrates the simplified system control architecture of the conventional COT control and the proposed dual-loop feedback AOT-controlled buck converter. The system is composed of a power stage and a feedback control stage. The power stage primarily includes a high-side and a low-side power transistor for switching and an LC filter that passes the energy from the input to the output.

In order to enhance the load transient response speed, a dual-loop feedback control stage is applied. A fast loop ensures a fast transient response speed, and a slow loop compensates for the output DC offset introduced by the output voltage step. The feedback loops are, as shown in Figure 1, the output voltage is divided by RF1 and RF2 to produce a feedback voltage VFB, which is the input of loop comparator COMP. The comparator controls the on-time generator to generate the Ton, which is the input of the logic and driver module to control the power transistors. This loop is the fast feedback loop, which, from the output voltage Vout directly to the comparator, without the need to go through the error amplifier (EA) and its compensation network, can achieve a fast response when the load transient step occurs. The presence of the fast feedback loop greatly improves the transient response speed of the system. However, the output voltage has a large DC detuning due to the limited gain of the loop. Therefore, another slow feedback loop is designed to amplify the feedback voltage VFB with a reference voltage VREF in the error amplifier first and then input to the loop comparator COMP to keep the DC value of the output voltage stable. Figure 2 shows the block diagram architecture of the proposed AOT buck converter.

Since the system sampled the output voltage in loop feedback control, the output voltage contains a capacitive voltage that is phase lagged compared to the inductor current. It may cause the output voltage ripple and inductor current ripple to be out of phase if this portion of the voltage is too large, which would make the system operating unstable [21,22]. The stability of the system was analyzed using the descriptive function method, with the addition of the perturbation $v_{c}\left(t\right)=r_{0}+\hat{r}*sin\left(2\pi f_{m}*t+\theta \right)$ at the control voltage Vc. Considering the output voltage ripple, Equation (1) was derived according to the law of modulation.
(1)$$v_{c}\cdot \left(t_{i-1}{+T}_{\mathrm{off}\left(i-1\right)}\right){+S}_{n}\cdot T_{\mathrm{on}}\;-\;\int_{t_{i-1}{+T}_{\mathrm{off}\left(i-1\right)}}^{t_{i}{+T}_{\mathrm{off}\left(i\right)}}\frac{i_{L}\left(t\right){\;-\;v}_{\mathrm{out}}\left(t\right){/R}_{L}}{C}\mathrm{dt}\;{=v}_{c}\cdot \left(t_{i}{+T}_{\mathrm{off}\left(i\right)}\right){+S}_{f}\cdot T_{\mathrm{off}\left(i\right)}$$
where $T_{\mathrm{off}\left(i\right)}$ is the turn-off time of the high-side transistor in the cycle i, and S_n_ and S_f_ are the rising and falling slopes of the voltage signal converted via inductor current.
(2)$$S_{n}=\frac{R_{i}\cdot \left(V_{\mathrm{in}}{\;-\;V}_{\mathrm{out}}\right)}{L}$$
(3)$$S_{f}=\frac{R_{i}\cdot V_{\mathrm{out}}}{L}$$

Suppose $T_{\mathrm{off}\left(i\right)}{=T}_{\mathrm{off}}+\Delta T_{\mathrm{off}\left(i\right)}$, where $T_{\mathrm{off}}$ is the off time of the high-side transistor at steady state, and $\Delta T_{\mathrm{off}\left(i\right)}$ is the perturbation of the high-side transistor off time at cycle i. Assuming that the steady state period is $T_{\mathrm{sw}}$, the beginning time of the cycle i can be expressed as
(4)$$t_{i}=\left(i\;-\;1\right)\cdot T_{\mathrm{sw}}+\sum_{k=1}^{i\;-\;1}\Delta T_{\mathrm{off}\left(k\right)}$$

Substitute Equation (4) into Equation (1) to obtain
(5)$$\begin{array}{c} S_{f}\cdot \left[\left(1+\frac{T_{\mathrm{off}}}{2\cdot C\cdot R_{\mathrm{ESR}}}\right)\cdot \Delta T_{i}\;-\;\left(1\;-\;\frac{{2T}_{\mathrm{on}}{+T}_{\mathrm{off}}}{2\cdot C\cdot R_{\mathrm{ESR}}}\right)\cdot \Delta T_{i-1}\right]\\ {=[v}_{c}\cdot \left(t_{i-1}{+T}_{\mathrm{off}\left(i-1\right)}\right){\;-\;v}_{c}\cdot \left(t_{i}{+T}_{\mathrm{off}\left(i\right)}\right){]\;-\;[v}_{c}\cdot \left(t_{i-2}{+T}_{\mathrm{off}\left(i-2\right)}\right){\;-\;v}_{c}\cdot \left(t_{i-1}{+T}_{\mathrm{off}\left(i-1\right)}\right)]\;-\;\\ \frac{v_{c}\cdot \left(t_{i-1}{+T}_{\mathrm{off}\left(i-1\right)}+\frac{\pi }{{2\pi f}_{m}\cdot 2}\right){\;-\;v}_{c}\cdot \left(t_{i}{+T}_{\mathrm{off}\left(i\right)}+\frac{\pi }{{2\pi f}_{m}\cdot 2}\right)}{{2\pi f}_{m}\cdot R_{L}C}+\\ \frac{v_{c}\cdot \left(t_{i-2}{+T}_{\mathrm{off}\left(i-2\right)}+\frac{\pi }{{2\pi f}_{m}\cdot 2}\right){\;-\;v}_{c}\cdot \left(t_{i-1}{+T}_{\mathrm{off}\left(i-1\right)}+\frac{\pi }{{2\pi f}_{m}\cdot 2}\right)}{{2\pi f}_{m}\cdot R_{L}C} \end{array}$$

Similarly, the perturbed duty cycle and the inductor current can be expressed as
(6)$$d\left(t\right)=\overset{M}{\underset{i=1}{\sum }}\left[u\cdot \left({t\;-\;t}_{i}{\;-\;T}_{\mathrm{off}\left(i\right)}\right)\;-\;u\cdot \left({t\;-\;t}_{i}{\;-\;T}_{\mathrm{off}\left(i\right)}{\;-\;T}_{\mathrm{on}}\right)\right]$$
(7)$$i_{L}\left(t\right){=i}_{L0}+\int_{0}^{i}\left[\frac{V_{\mathrm{in}}}{L}\cdot d\left(t\right)\;-\;\frac{V_{\mathrm{out}}}{L}\right]\cdot \mathrm{dt}$$

Fourier analysis of the inductive current and substitution into Equation (1) gives
(8)$$c_{m}=j\cdot \frac{{2f}_{m}}{N}\overset{M}{\underset{i=1}{\sum }}\int_{t_{i}{+T}_{\mathrm{off}\left(i\right)}}^{t_{i}{+T}_{\mathrm{off}\left(i\right)}{+T}_{\mathrm{on}}}e^{-j\cdot {2\pi f}_{m}\cdot t}\cdot \mathrm{dt}\;\approx \;\frac{f_{s}}{S_{f}}\cdot \frac{\left({1-e}^{-j\cdot {2\pi f}_{m}\cdot T_{\mathrm{on}}}\right)\left({1\;-\;e}^{-j\cdot {2\pi f}_{m}\cdot T_{\mathrm{sw}}}\right)\left(1-\frac{e^{j\pi /2}}{{2\pi f}_{m}\cdot R_{L}\cdot C}\right)}{\left(1+\frac{T_{\mathrm{off}}}{2\cdot C\cdot R_{\mathrm{ESR}}}\right)\;-\;\left(1\;-\;\frac{{2T}_{\mathrm{on}}{+T}_{\mathrm{off}}}{2\cdot C\cdot R_{\mathrm{ESR}}}\right)e^{-j\cdot {2\pi f}_{m}\cdot T_{\mathrm{sw}}}}\cdot \frac{V_{\mathrm{in}}\cdot e^{-j\theta }}{j\cdot {2\pi f}_{m}\cdot L}$$

According to the equation of the control voltage, the transfer function from the control signal V_c_ to the inductor current in the s domain can be obtained as
(9)$$\frac{i_{L}\left(s\right)}{v_{c}\left(s\right)}=\frac{f_{s}}{s_{f}}\cdot \frac{\left({1\;-\;e}^{-s\cdot T_{\mathrm{on}}}\right)\left({1\;-\;e}^{-s\cdot T_{\mathrm{sw}}}\right)}{\left(1+\frac{T_{\mathrm{off}}}{2\cdot C\cdot R_{\mathrm{ESR}}}\right)\;-\;\left(1\;-\;\frac{{2T}_{\mathrm{on}}{+T}_{\mathrm{off}}}{2\cdot C\cdot R_{\mathrm{ESR}}}\right)e^{-s\cdot T_{\mathrm{sw}}}}\cdot \frac{V_{\mathrm{in}}}{s\cdot L}$$

The transfer function from the output voltage V_out_ to the control voltage V_c_ is obtained.
(10)$$\frac{V_{\mathrm{out}}\left(s\right)}{v_{c}\left(s\right)}=\frac{i_{L}\left(s\right)}{v_{c}\left(s\right)}\cdot \frac{R_{L}{(R}_{\mathrm{ESR}}C\cdot s+1)}{\left(R_{L}{+R}_{\mathrm{ESR}}\right)C\cdot s+1}$$

When $R_{L}\gg R_{\mathrm{ESR}}$, the transfer function is
(11)$$\frac{v_{\mathrm{out}}\left(s\right)}{v_{c}\left(s\right)}\;\approx \;\frac{f_{s}}{s_{f}}\cdot \frac{V_{\mathrm{in}}}{\mathrm{sL}}\cdot \frac{1}{1+\frac{s}{Q_{1}\omega_{1}}+\frac{s^{2}}{\omega_{1}^{2}}}\cdot \frac{\left(R_{\mathrm{ESR}}\cdot C\cdot s+1\right)}{1+\frac{s}{Q_{2}\omega_{2}}+\frac{s^{2}}{\omega_{2}^{2}}}$$
where $Q_{1}=2/\pi$, $\omega_{1}{=\pi /T}_{\mathrm{on}}$ and $Q_{2}=\;T_{\mathrm{sw}}/\left(R_{\mathrm{ESR}}\cdot {C\;–\;T}_{\mathrm{on}}/2\right)\cdot \pi$, $\omega_{2}{=\pi /T}_{\mathrm{sw}}$.

As can be seen from the formula, Q_2_ will vary with the output capacitance Co, equivalent series resistance (ESR), conduction time, and switching period, which have the possibility of splitting the poles of the right half plane. Therefore, the loop stability condition is
(12)$$R_{\mathrm{ESR}}\cdot C\;>\;\frac{\;T_{\mathrm{on}}}{2}$$

In addition, Q_2_ varies sensitivity with different applications and peripheral devices. To ensure loop stability, the TYPE-I compensation is selected, which introduces only one main pole, and the low-frequency real poles in extreme cases should be placed outside the gain-bandwidth product (GBW) of the feedback loop [23,24]. The loop model under this compensation mode in Simplis is established, and the actual values of external components are substituted. The Simplis simulation of the loop stability is shown in Figure 3. As can be seen in the figure, in this compensation mode, the feedback loop can remain stable.

## 3. Circuit Implementations

### 3.1. AOT Generator

When the COT-controlled buck converter operates under the steady state of PWM mode without the consideration of the parasitic parameters or conversion efficiency, the on time of the high-side transistor can be formulated as
(13)$$T_{\mathrm{on}}=D\cdot T_{\mathrm{sw}}=\frac{V_{\mathrm{out}}}{V_{\mathrm{in}}}\cdot \frac{1}{f_{\mathrm{sw}}}$$

The above equation shows that the operating frequency of the converter is related to the input and output voltages with a constant on-time Ton of the transistor. In order to obtain a fixed operating frequency, it is necessary to dynamically adjust the on time at different input and output voltage [25,26].

The structure of the adaptive on-time generator used in this paper is shown in Figure 4, which consists of three main parts: V-I converter, a capacitor charging and discharging circuit, and a voltage comparator.

Resistors R1~R3, operational amplifier, and MN1 constitute the V-I converter. R1 and R2 are used to divide the voltage while isolating the high voltage at Vin. The operational amplifier makes the voltage on R3 equal to the divided voltage to determine the charge current, which is shown in Equation (14). This current is then mirrored to MP2 to finally charge the capacitor.
(14)$$I_{c}{=V}_{\mathrm{in}}\cdot \frac{R2}{R1+R2}\cdot \frac{1}{R3}$$

Since the range of Vout is close to the supply voltage, it is also necessary to divide Vout before comparison. The capacitor is charged until the voltage is equal to the division of Vout, the comparator flips, and the on time of the high-side transistor ends. At the same time, MN2 turns on and discharges the capacitor in preparation for the next charging cycle. Therefore, without consideration of the delay, the adaptive on-time of the generator can be expressed as
(15)$$T_{\mathrm{on}}=\frac{C}{I_{c}}\cdot \frac{R5}{R4+R5}\cdot V_{\mathrm{out}}=\frac{C\cdot R3\cdot R5\cdot \left(R1+R2\right)}{R2\cdot \left(R4+R5\right)}\cdot \frac{V_{\mathrm{out}}}{V_{\mathrm{in}}}$$

So, the on time would vary with changes in input voltage and output voltage at the stable state of this buck converter. Substituting Equation (14) into Equation (15), we can see the switching frequency is now constant:(16)$$f_{\mathrm{sw}}=\frac{R2\cdot \left(R4+R5\right)}{C\cdot R3\cdot R5\cdot \left(R1+R2\right)}$$

### 3.2. Zero Current Detection

When the buck converter works under a light load, the inductor current may drop to zero at each cycle, which is shown in Figure 5b. If the inductor current drops below 0 A, there will be a reverse current going through the body diode of the power transistor, causing a reduction in efficiency [27]. In order to improve efficiency under light loads, the buck converter proposed in this paper could operate in PSM under light load, and it turns off the low-side transistor once the inductor current drops to zero and keeps the inductor current at zero until the energy stored in the output capacitor is depleted [28,29].

To improve the response speed and reduce unnecessary losses, an accurate inductor zero-current detection circuit is applied [30]. Since the on resistance of the power transistor and the voltage change at node SW is relatively small, the comparator needs to work under negative input. Figure 6 shows the circuit of the inductor zero-current detection comparator designed in this paper. It comprises a three-stage differential amplifier. Since the SW voltage variation is small and may be negative, the common-mode input range needs to contain zero voltage, which also needs to reduce noise and improve sensitivity at the same time. So, the BJT emitter input and collector output are used to sensitively amplify the error between the ZCD_SW and ZCD_REF. The switching speed of these zero current detection comparators mainly depends on the delay of the amplifiers and the inverter [31].

## 4. Experimental Results

The proposed fast transient AOT-controlled buck converter was implemented and fabricated in the TSMC 0.18 µm BCD process. Figure 7 shows the chip microphotograph. The chip size is 3000 µm × 3000 µm, which includes the integrated power transistors, PAD, and electrostatic discharge (ESD) protection circuit. The input voltage range is 5.5 to 15 V, the output voltage range is 0.5 to 5 V, and the output load current range is up to 5 A.

The steady-state PWM operating waveform of the system is illuminated in Figure 8. The input voltage is 15 V, the output voltage is 5 V, and the load current is 1 A. As shown in Figure 8, the converter works normally under PWM and is capable of driving the output load. As can be seen from Figure 8, the converter works smoothly in PWM mode, and there is no large overshoot and oscillation. The measured output voltage is about 4.89 V, and the output voltage ripple is about 11 mV. The overshoot and undershoot of the VSW and VBST is significantly less than 1 V. The steady operating frequency is about 2.17 MHz.

As the load current decreases, the converter could work in the PSM mode, and Figure 9 shows the operating waveform under steady-state PSM, with an input voltage that is 15 V, an output voltage of 5 V, and a load current of 10 mA. As can be seen from the figure, under light loads, the converter reduced frequency and operated in PSM mode, the inductor current waveform was in the form of a single pulse, and there was no sub-harmonic oscillation, showing that the error amplifier and its compensation network, PWM comparators, etc., could operate stable. The measured output voltage is about 4.71 V, and the output voltage ripple is about 24 mV.

Figure 10 shows the load transient response waveform when the load current was changed from 1 A to 200 mA and from 200 mA to 1 A. As can be seen from the figure, when the load was changed transiently, the VOUT returned quickly to the stable state without sub-harmonic oscillation and burrs. The stable value is around 4.9 V. When the load current was changed from 1 A to 200 mA, the overshoot was about 57 mV, and the recovery time was about 15 μs. When the load was changed from 200 mA to 1 A, the undershoot was about 53 mV, and the recovery time was about 13 μs.

The converter frequency changes with input and output voltages are shown in Figure 11. Due to the adaptive on-time generator proposed in this paper, the switching frequency can be maintained at a quasi-fixed value over a large input and output range. The test results show that in the input voltage range from 5 V to 15 V, the frequency changes can be controlled within 9%. In the output voltage from 0.5 V to 4.5 V, the frequency changes can be controlled within 14%.

The converter efficiency is shown in Figure 12. The input voltage and the output voltage are 7.5 V and 2.5 V, respectively, and the load current was set to 100 mA, 500 mA, 1 A, 2 A, and 5 A. The peak efficiency is 85.14% at 2 A load current. As the load current decreases, the system efficiency decreases because of the decay of the output power, but the system loss almost does not change. Due to the accurate zero current detection and PSM control used at light load, at a load current of 100 mA, the system efficiency only dropped to 71%. If the converter is forced to operate in PWM mode at light load, the efficiency will drop to 19% at the load current of 100 mA. And with the increasing of load current, the conduction loss of power transistors rises, which means the conversion efficiency gradually decreases. Under the maximum load of 5 A, the conversion efficiency dropped to about 83.15%.

The main performance metrics of the proposed buck converter are summarized and compared with other recently published buck converters in Table 1. References [32,33] are conventional current mode control. References [34,35] are the COT control same as this paper. It can be seen that the dual modulation adaptive-on-time controlled mode buck converter designed in this paper achieves a better response speed than both the current mode controlled and the conventional constant-on-time controlled converters compared over a wide input and output range, significantly, which keeps the upper and lower overshoot voltage within 60 mV with the load transient response recovery time of about 13 μs and 15 μs.

## 5. Conclusions

A fully integrated AOT-controlled buck converter with fast load transient response speed is proposed in this paper. The converter was implemented in the TSMC 0.18 μm BCD process, and a fast and stable load transient response was achieved due to the dual-loop feedback control. The proposed circuit solved the problem caused by the frequency variations in the traditional COT method via an adaptive on-time generator. The converter could operate in PSM under light load to improve light load efficiency via the zero-current detection circuit. Efficiency can still be improved under heavy loads. The experimental results showed that based on the dual modulation adaptive-on-time controlled mode, the system could operate stably under different conduction modes and transform without inducing large undershoot voltage and response time. The rising and falling load transient response recovery time was approximately 13 μs and 15 μs, respectively, with the overshoot and undershoot voltages kept below 60 mV. The improved load transient response time and improved power efficiency under light load make the proposed buck converter a promising candidate for microprocessors.

## Figures and Tables

**Figure 1 micromachines-14-01868-f001:**
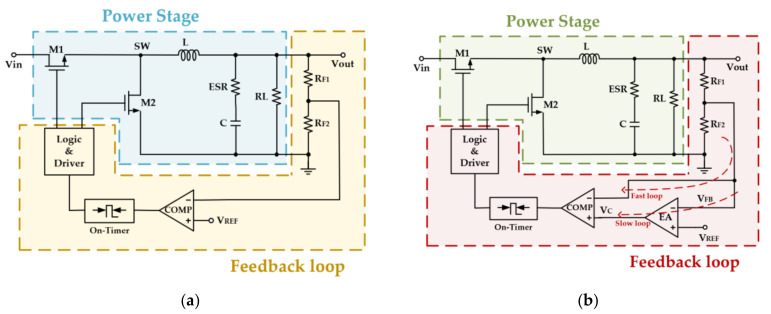
System control architecture of the AOT buck converter: (**a**) Conventional COT control; (**b**) the dual-loop feedback control applied in this paper.

**Figure 2 micromachines-14-01868-f002:**
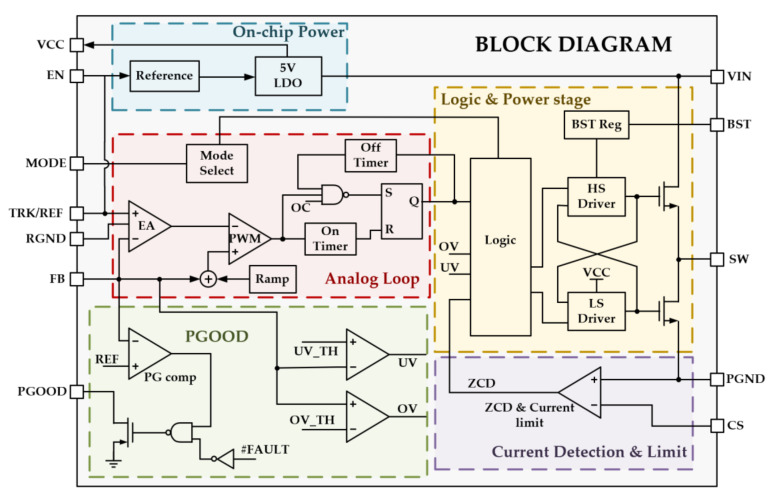
Block diagram of the proposed AOT buck converter.

**Figure 3 micromachines-14-01868-f003:**
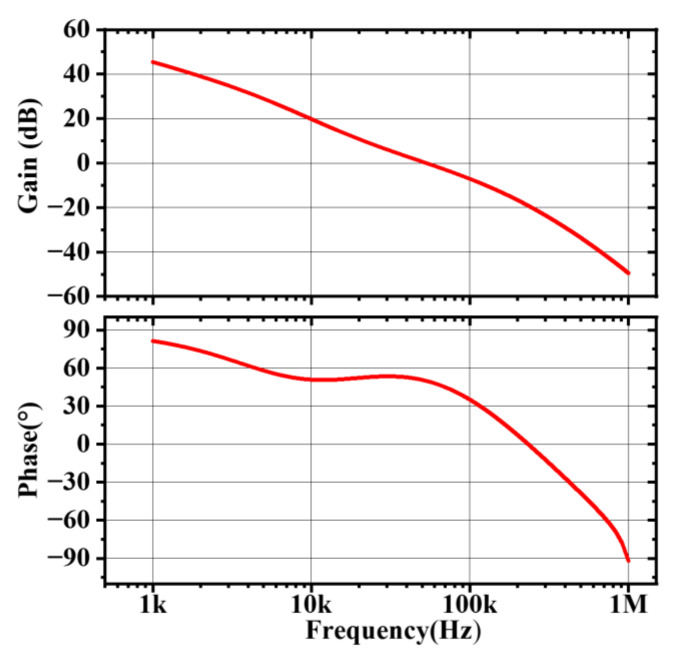
Bode plot of the system.

**Figure 4 micromachines-14-01868-f004:**
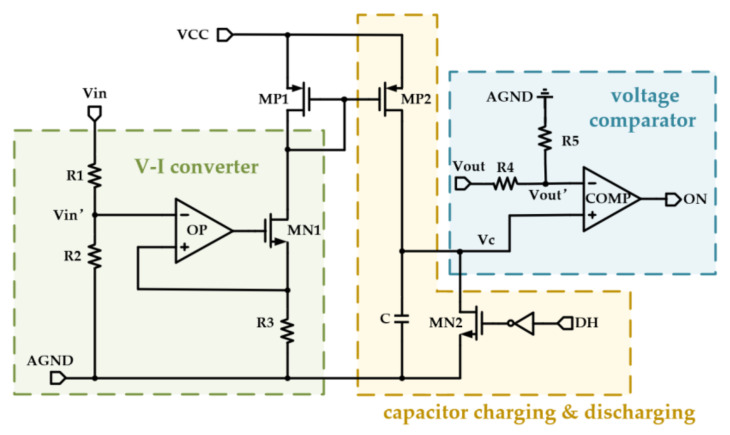
The structure of the adaptive on-time generator.

**Figure 5 micromachines-14-01868-f005:**
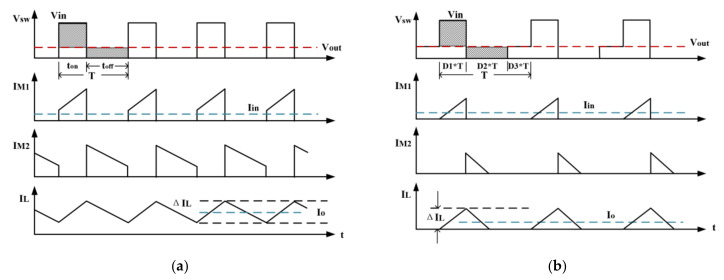
Operating waveforms of the BUCK converter: (**a**) heavy load operating and (**b**) light load operating.

**Figure 6 micromachines-14-01868-f006:**
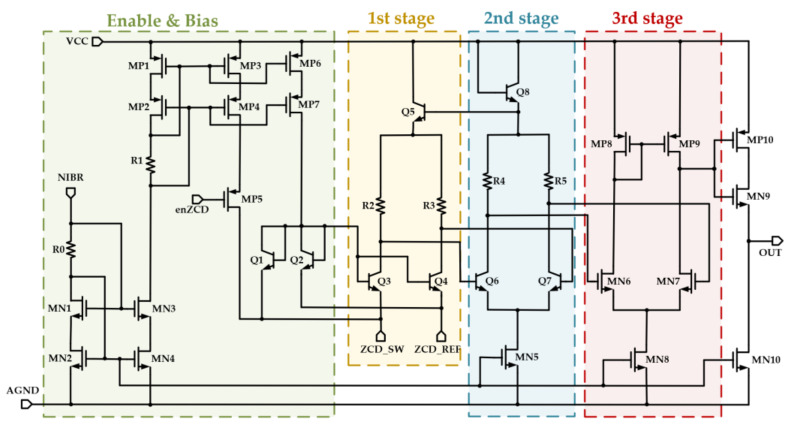
Inductor zero current detection comparator circuit.

**Figure 7 micromachines-14-01868-f007:**
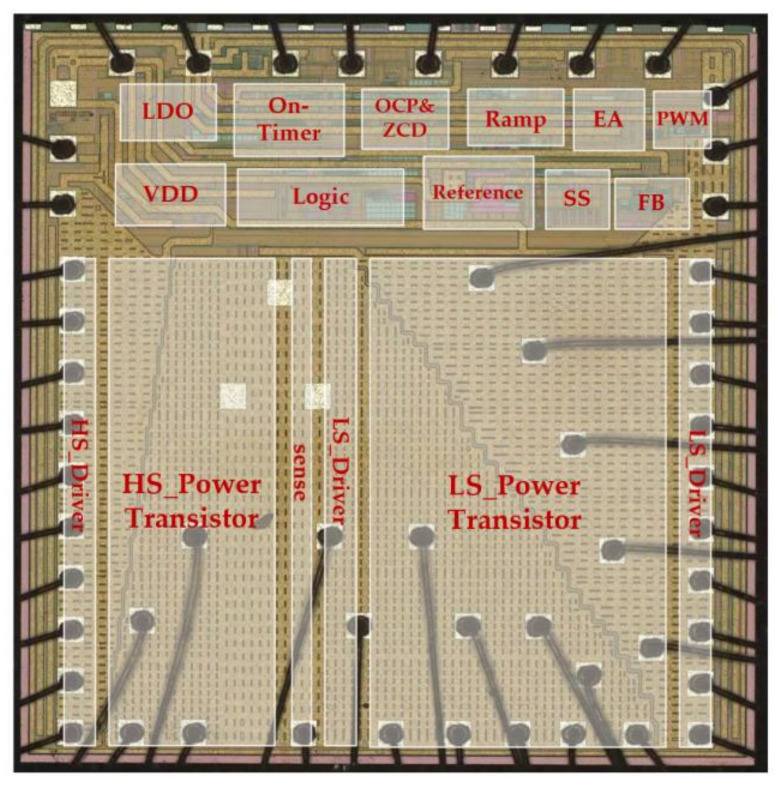
The chip microphotograph.

**Figure 8 micromachines-14-01868-f008:**
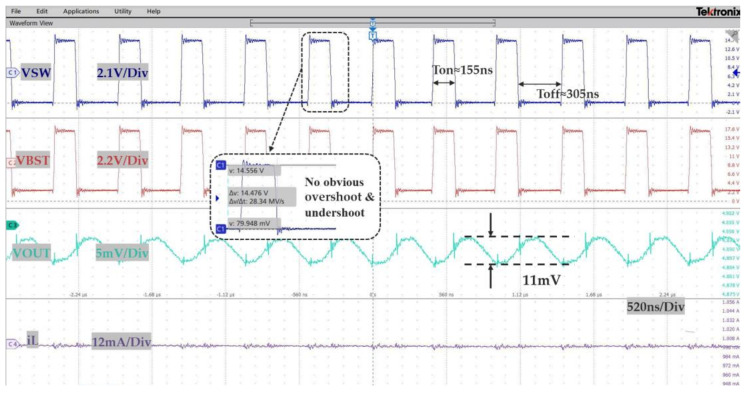
The steadystate PWM operating waveform.

**Figure 9 micromachines-14-01868-f009:**
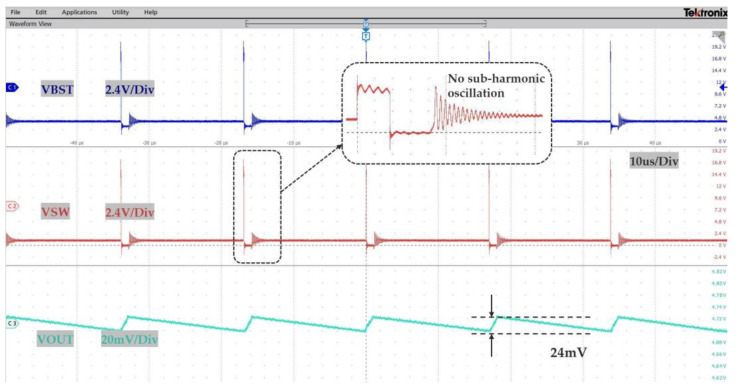
The steady-state PSM operating waveform.

**Figure 10 micromachines-14-01868-f010:**
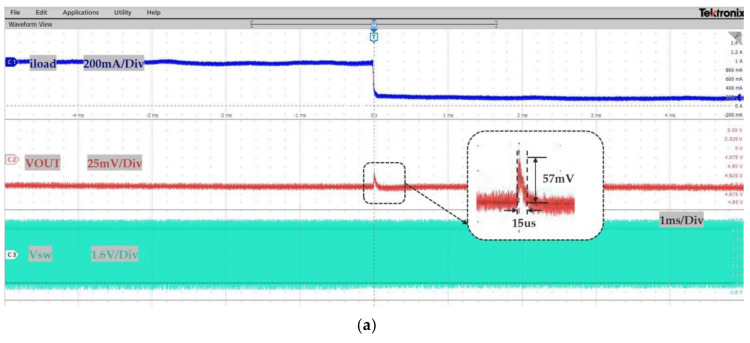
The load transient response waveform: (**a**) load current falling and (**b**) load current rising.

**Figure 11 micromachines-14-01868-f011:**
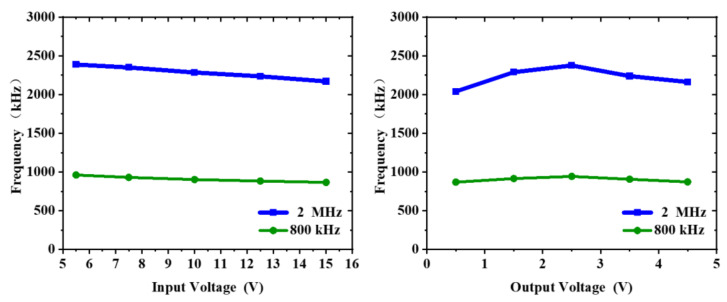
The frequency varied with input and output voltage.

**Figure 12 micromachines-14-01868-f012:**
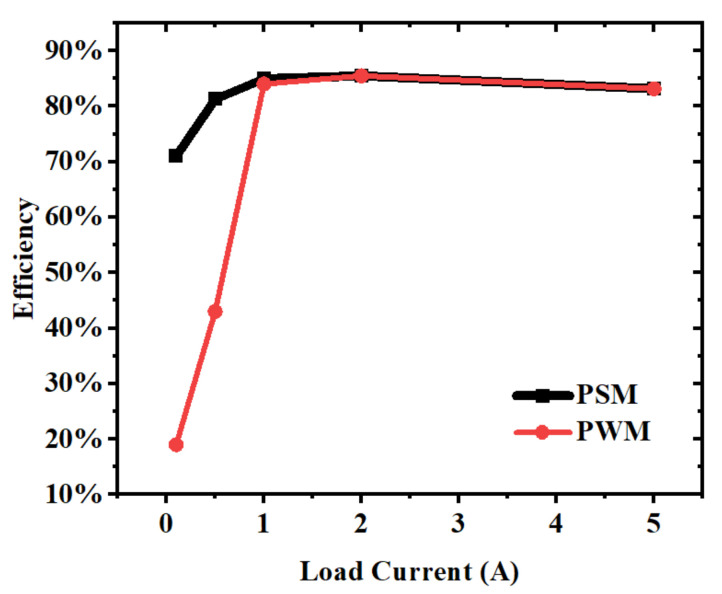
Efficiency at different load currents.

**Table 1 micromachines-14-01868-t001:** Main performance summary and comparison.

Reference	[32]	[33]	[34]	[35]	This Work
**Process**	0.5 μm	0.25 μm	0.18 μm	0.18 μm	**0.18 μm**
**Operation Mode**	Current mode	Current mode	ACOT	RBCOT	**ACOT**
**Input Voltage Range [V]**	2.5~5.5	3~4.5	2.1~5.5	3.3	**5.5~15**
**Output Voltage Range [V]**	0.9~5	1.8	1.8	1.03	**0.5~5**
**Inductor [μH]**	1.8	1	2.2	10	**4.7**
**Output Capacitor [μF]**	100	4.7	10	10	**47**
**Load Range [A]**	0~5	0~5	0~200 m	0~1	**0~5**
**Switching Frequency**	1 MHz	5 MHz	-	-	**2 MHz**
**Output ripple [mV]**	<10	50	19	18	**11**
**Recovery time (rise) [μs]**	20	14	30	18	**13**
**Recovery time (fall) [μs]**	50	15	24	30	**15**
**Overshoot [mV]**	60	60	48	330	**57**
**Undershoot [mV]**	90	60	43	200	**53**

## Data Availability

Not applicable.

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
