# Peer review of "A Fast Transient Adaptive On-Time Controlled BUCK Converter with Dual Modulation"

_micromachines, 2023, doi:10.3390/mi14101868_

Round 1

Reviewer 1 Report

Here are my comments about the study:

1) In the introduction section, the advantages of the proposed approach and its differentiation from other techniques in the literature should be emphasized.

2) The originality should be highlighted throughout the article.

3) Results obtained from figures and tables should be thoroughly interpreted and discussed.

4) The introduction section should incorporate and discuss publications related to buck converter designs, such as 

https://doi.org/10.1016/j.jfranklin.2016.04.008

https://doi.org/10.1016/j.aej.2021.07.037

https://doi.org/10.1016/j.sigpro.2006.02.022

https://doi.org/10.1016/j.isatra.2012.07.005 

https://doi.org/10.1016/j.prime.2022.100039

https://doi.org/10.1016/j.ijepes.2022.108099 

5) Utilizing the Bode diagram in Figure 3, the frequency response of the relevant system should be interpreted.

6) Care should be taken with equation formulations, and the quality of figures should be improved.

Moderate editing of English language required.

Reviewer 2 Report

In this paper, authors proposed a fast adaptive on-time controlled buck converter with a dual feedback control loop, the aim was to ensure a high transient response speed without affecting the stability of the system, experimental results are also presented. The paper is well written and well presented. I have some questions and some suggestions:

1) Equation 13 is true only in the steady state, and when the efficiency of the system is near 100%. Can you give more details about this assumption.

2) For the experimental part, it would be better to add a photo of the whole test bench (DC source, load, oscilloscope …etc.).

3) Future works are missed in the conclusion section.   

The paper is well written.

Reviewer 3 Report

Dear colleagues, the article seems good, but I still have some improvement suggestions.

You can improve the introduction and some explanations by sticking to the following topics. It is true that in the COT control, the duty cycle is fixed for a defined input and output voltage, but this also happens in the traditional control, which is not related to the fact that the on-time of power transistors per cycle is kept constant in COT control. Actually, the duty cycle vs. gains has not changed, but the duty cycle changes at the same time as the switching frequency. Could you explain why the output voltage ripple introduces a DC offset?

Some paragraphs seem to come from: “A high performance adaptive on-time controlled valley-current-mode DC–DC buck converter”. Journal of Semiconductors, 2020, 41 (6): 062402, Published Online: Sep. 10, 2021. But it seems you didn´t cite the article, please do it.

Please include more comments about the switching ripple in the inductor and the capacitor. It seems that if you decrease the switching frequency under light load conditions, the switching ripples will increase. I am wondering if this may affect any design specification or maybe passive components would need to be larger to achieve the same performance. I am not sure about this; just want to know your comments related to this topic.

The English grammar is ok. There are a few improvements. A couple of examples are listed here (the English revision is not exhaustive).

On page 1, line 14, it says:

“as load current decreases”

- I suggest changing it to:

“as the load current decreases”

On page 1, line 27, it says:

“power supply modules have played an increasing critical role in the system”

- I suggest changing it to:

“power supply modules have played an increasingly critical role in the system”

On page 1, line 36, it says:

“increasing demand of intelligence”

- I suggest changing it to:

“increasing demand for intelligence”

On page 1, line 37, it says:

“higher demands on dynamic characteristics”

- I suggest changing it to:

“higher demands on the dynamic characteristics”

On page 1, line 38, it says:

“the output of converter to keep stable under different load conditions”

- I suggest changing it to:

“the converter's output is stable under different load conditions”

On page 1, line 41, it says:

“but have more complex compensation circuits”

- I suggest changing it to:

“but has more complex compensation circuits”

On page 2, line 55, it says:

“pulse width modulation (PWM) combined with pulse skip modulation (PSM) were used in this paper [14]”

- I suggest changing it to:

“pulse width modulation (PWM) combined with pulse skip modulation (PSM) has been used [14]”

On page 2, line 65, it says:

“as well as ensure system stability”

- I suggest changing it to:

“as well as ensures system stability”

On page 2, line 66, it says:

“This converter could work under PWM control, which keep the operating frequency stable, and reduce”

- I suggest changing it to:

“This converter could work under PWM control, which keeps the operating frequency stable and reduces”

Round 2

Reviewer 1 Report

The authors have responded quite reasonably to the reviewers' comments. However, before the article is published in the journal, attention should be paid to the journal's writing guidelines, the format of references, and the language used in the article.

Minor editing of English language required.

Reviewer 3 Report

My comments have been addressed. Thank you.

My comments have been addressed. Thank you.